# EMERGENT CORPUS PRE-TRAINING BENEFITS VISION LANGUAGE MODELING

## ABSTRACT

Vision Language Pre-trained Models (VL-PTMs) have achieved state-of-the-art results across various tasks, but their effectiveness heavily relies on large-scale multimodal datasets. While VL-PTMs excel in scenarios with abundant data, they struggle to achieve sample efficiency in tasks with limited data resources. In this work, we explore the use of Emergent Communication (EC) for knowledge transfer in VL-PTMs. In particular, we pre-train a state-of-the-art Vision Language (VL) model on a corpus of EC tokens, generated through a referential game involving two artificial agents. Through experiments on three diverse cross-modal matching and reasoning tasks, we demonstrate significant performance improvements. For instance, EC pretraining enhances Visual Referring Expression (VRE) accuracy by 108.6% while improving Visual Entailment (VE) performance by 69.6%. We further demonstrate that a vision-language model, exclusively pre-trained on EC tokens from scratch utilizing a sequence-to-sequence learning objective, can be effectively leveraged for fine-tuning numerous other vision-language downstream tasks, outperforming baseline settings without any pretraining and in some cases significantly narrowing the performance gap with models pre-trained on natural language. These results highlight the transferability and generalization capabilities of EC pretraining across different VL tasks and the potential of leveraging the multimodal grounding of EC tokens to enhance VL understanding in resource-constrained settings, especially in settings with limited natural language data. We discuss implications and propose avenues for future research to explore the connections between EC and VL for multimodal understanding and effective human-machine communication.

## 1 INTRODUCTION

Over the years, significant advancements have been made in Vision Language Pre-trained Models (VL-PTMs). The prevailing approach has been to build larger models, incorporating more parameters in order to enhance their generalizability Du et al. (2022). Alternative efforts have been directed toward devising improved architectures for the fusion of text and image representations, such as fusion encoders, dual encoders, or a combination thereof Radford et al. (2021); Li et al. (2020a). Despite these remarkable advancements, the challenge of achieving robust generalization in low-resource scenarios persists for VL-PTMs. A pivotal factor contributing to this limitation is the scarcity of parallel data for numerous real-world cross-modal applications and tasks, including but not limited to image-text retrieval, phrase grounding, visual question answering, and video localization Xu et al. (2023); Li et al. (2023); Liu et al. (2023a).

Self-supervised Learning (SSL) has emerged as a leading strategy to tackle challenges related to limited resources and label scarcity in vision-language learning. By employing contrastive learning and pretext tasks, SSL has facilitated substantial progress, exemplified by models such as CLIP Radford et al. (2021) and VisualBERT Li et al. (2019) that can effectively learn cross-modal representations. Despite these achievements, SSL methods encounter several challenges that can limit practical utility, *e.g.*, task design dependency, computational complexity, sensitivity to the choice of hyper-parameters, and catastrophic forgetting Purushwalkam et al. (2022); Tian et al. (2020); Ericsson et al. (2021). The study of Emergent Communication (EC) is a promising direction as an alternative paradigm that can address these challenges.

Emergent communication in multi-agent environments investigates the interactive and functional language ability of intelligent agents, for example, how deep agents collaboratively invent a shared language for task completion. This involves experimental validation on environments in which agents are trained to make sequences of decisions. Typically, agents are assigned a task to achieve and are also equipped with a shared reward function. In most studies, agents must learn to maximize the expected cumulative reward, which serves as the shared supervision signal Lazaridou et al. (2017; 2018); Havrylov & Titov (2017). Effective communication plays a pivotal role in facilitating collaborative task completion. A common approach to modeling EC in literature is to allow agents (usually two) to interact by playing a referential game Evtimova et al. (2018); Lazaridou et al. (2018); Li et al. (2020b); Yao et al. (2022). In this game, one agent, referred to as the Speaker Agent, is shown an input image and must generate a message, in the form of a discrete set of tokens that describes the image. The generated message is passed to another agent, known as the Listener Agent, who must select the correct input image from a set of distractors.

In the context of vision-language pretraining, EC offers several benefits. EC provides a practical alternative to natural language, particularly beneficial in resource-constrained settings. Recent works show that EC shares several common properties with natural language, such as compositionality, learning, structure, and symbolic representation Yao et al. (2022); Li et al. (2020b). These similarities motivates us to explore EC pretraining. If emergent communication proves effective for VLM-PTMs, it could be a valuable tool for training models when large labeled parallel vision-language datasets are scarce or expensive to obtain. The emergent messages encapsulate both visual and linguistic stimuli, providing a rich learning signal for vision-language modeling, enhancing the model's proficiency in understanding and generating meaningful associations between visual and linguistic elements.

Drawing on past successes demonstrating the benefits of emergent communication in language modeling Li et al. (2020b), in this paper, we extend the applicability of emergent language to vision-language learning, emphasizing its broader utility and effectiveness in multimodal contexts. Specifically, we explore the transferability of knowledge gained through EC pretraining across multiple vision-language downstream tasks. We present a series of experiments aimed at pretraining VLMs on a corpus of EC tokens and subsequently fine-tuning them for various downstream tasks, including cross-modal matching (*e.g.*, Visual Referring Expression) and cross-modal reasoning (*e.g.*, Visual Question Answering and Visual Entailment). Investigating the impact of EC on vision-language learning can improve our understanding on how models autonomously develop communication strategies, potentially providing insights into the inner workings of VLMs.

Our objective is to investigate the extent to which incorporating emergent language modeling confers transferable benefits to various downstream VL tasks. Through a comprehensive series of experiments, we observe significant improvements in downstream task performance. Specifically, EC pretraining improves VRE accuracy by $108.6\%$ compared to the baseline, with the model accuracy doubling from $29.81\%$ to $62.17\%$ on the validation set after pretraining on EC, showcasing the model's enhanced understanding of the visual context. For Visual Entailment, a task requiring complex reasoning, EC pretraining achieves a significant accuracy gain of $69.6\%$, highlighting its ability to capture nuanced semantic relationships between visual content and textual hypotheses. These findings open up new possibilities for leveraging EC to improve performance in other cross-modal tasks. Our results demonstrate the potential of EC corpus pretraining in VL applications.

The contributions of our work can be summarized as follows: **(1)** We introduce a learning framework (EC-VL) that leverages the interactive and functional language abilities of communicating agents to explore the benefits of emergent language in vision language model pretraining. We evaluate the effectiveness of EC pretraining on various VL downstream tasks, including Visual Referring Expression (VRE), Visual Question Answering (VQA), and Visual Entailment (VE). By considering a diverse range of tasks, we demonstrate the transferability and generalization capabilities of EC pretraining across different reasoning scenarios. **(2)** We provide empirical evidence of the transferability of benefits of EC pretraining in VL applications. Our results reveal significant performance gains across all evaluated downstream tasks, highlighting how incorporating emergent language modeling in pretraining substantially enhances the model's understanding and reasoning capabilities in various VL tasks. These findings open up new possibilities for leveraging EC to improve performance in other cross-modal tasks. **(3)** We provide insights into the nature of emergent language and its potential applications in VL-PTMs. We discuss the implications of our findings and propose potential extensions for future research to delve into the intricate connections between EC and VL for multimodal understanding and for effective human-machine communication.

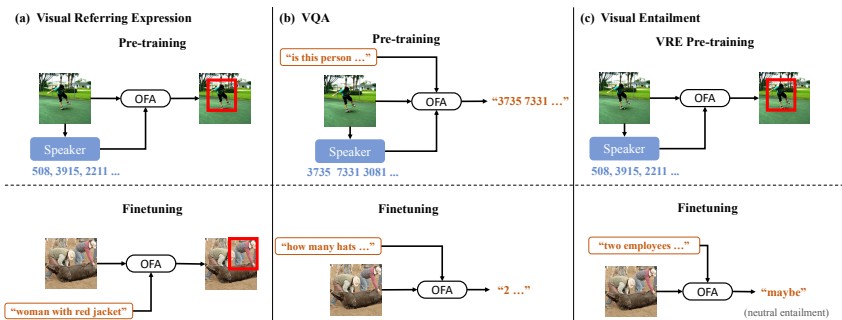

Figure 1: Method overview: a) VRE pretraining uses EC tokens for learning instead of natural language. VRE fine-tuning trains with natural language input. b) VQA pretraining uses EC text as a target answer while fine-tuning is performed on natural language answers. c) Visual Entailment adopts the VRE pre-trained model for fine-tuning in order to explore EC pretraining transfer.

## 2 RELATED WORK

**Emergent Communication.** Recently, there has been a growing interest in investigating the emergence of language within deep agent networks for task completion Lazaridou et al. (2017; 2018); Lazaridou & Baroni (2020); Mordatch & Abbeel (2018); Raviv et al. (2019a); Das et al. (2019). Pioneering works simulate communication while training agents Lazaridou et al. (2017); Evtimova et al. (2018). Building upon these foundations, Mordatch & Abbeel (2018); Raviv et al. (2019b) further explored the compositionality aspect of the emerged language. Several researchers have also attempted to interpret the structure and linguistic properties of the emergent language. For instance, Chaabouni et al. (2019) discovered that networks develop an anti-efficient encoding scheme, where longer messages are associated with the most frequent inputs, contrary to human language where the most frequent words are usually represented by shorter strings. Additionally, Patel et al. (2021) observed the emergence of egocentric grounded messages when agents were initialized in complex environments. In our work, instead of analyzing or explaining the emerging language, we focus on harnessing the inductive bias provided by the emergent language to enhance cross-modal vision and language learning. To this end, we explore the benefits of Emergent Communication (EC) for Vision Language pre-training and, interestingly, we gain valuable insights that contribute to a better understanding of EC language dynamics.

**Corpus Transfer of Synthetic or Emergent Communication Language.** While significant progress has been made in analyzing the properties of emergent communication language Mordatch & Abbeel (2018); Resnick et al. (2020); Lazaridou & Baroni (2020); Chaabouni et al. (2019), a different line of research has explored the potential transfer benefits of synthetic and emergent language in improving learning and generalization in language models Yao et al. (2022); Li et al. (2020b); Papadimitriou & Jurafsky (2020). For instance, Papadimitriou & Jurafsky (2020) investigated the structural awareness of language models, measuring how much the language structure acts as an inductive bias to enhance learning when transferring from one language or structure to another. They trained models on data with varying degrees of language-like structure, such as music, Java code, and nested symbols, and evaluated their performance on natural language to examine the presence of an inductive bias. Similarly, Yao et al. (2022) linked emergent languages and natural languages by investigating whether modeling an emergent language provides transferable benefits for downstream natural language tasks, including language modeling and image captioning. Their results demonstrated that pre-training on emergent communication text can achieve comparable performance to models pre-trained on natural language, particularly in low-resource settings. Furthermore, Li et al. (2020b) investigated the potential of an emergent communication protocol pre-trained without any human language data to benefit downstream NLP applications like machine translation. Their findings across various language pairs and few-shot setups indicate that parameter initializations informed by EC pre-training, along with adapter modules and annealed regularization, yield higher accuracy and improve sample efficiency compared to baselines without EC pertaining. We enrich this body of work by investigating the benefits of EC pertaining to Vision Language learning. We argue that by leveraging the Emergent Corpus text, generated through a referential game grounded in both

vision and language, we can maximize its utility in cross-modal settings. We experimentally show that EC corpus transfer can enhance vision language tasks.

**Pre-training for Vision Language Transfer Learning.** In the field of cross-modal machine learning, the predominant approach has been to pre-train VLMs on multimodal data and subsequently fine-tune them for various downstream tasks Wang et al. (2022a); Gan et al. (2022); Du et al. (2022). Tasks involving reasoning and grounding, such as Visual Question Answering Antol et al. (2015), Phrase grounding Yu et al. (2016), as well as Image Captioning Lin et al. (2014), have greatly benefited from these pre-trained models, leading to significant performance improvements Zellers et al. (2019); Cho et al. (2021). Several successful architectures have been proposed to model visual language interactions Li et al. (2020a; 2019). For instance, Fusion Encoders Li et al. (2019); Su et al. (2020) employ self-attention or cross-attention operations on text embeddings and image features, while Dual Encoders Radford et al. (2021); Wang et al. (2022b) utilize separate encoders for each modality, making them highly efficient for tasks like image-text retrieval Du et al. (2022); Lu et al. (2019). Different pre-training objectives also exist for various vision language downstream tasks. One common approach is Cross-Modal Masked Language Modeling (MLM) Liu et al. (2023b); Kim et al. (2021). In cross-modal MLM, the Vision Language pre-trained model leverages unmasked tokens and image features to predict masked tokens. Another approach is Cross-Modal Masked Region Prediction (MRP), where Regions of Interest (RoIs) are masked and predicted based on other image features Chen et al. (2020). Recent works have explored pre-training using image-text matching objectives, akin to next sentence prediction (NSP) tasks in NLP Huang et al. (2021), or Cross-Modal Contrastive Learning (CMCL) that aims to learn universal vision and language representations within the same semantic space Li et al. (2021). CMCL achieves this by encouraging matched image-text pairs to have similar embeddings while pushing non-matched embeddings further apart. While numerous studies have focused on scaling up models to billions of parameters, designing better VL model architectures, and developing optimal pre-training tasks, in this work we address fundamental questions on the potential of EC as a powerful paradigm for pre-training vision language tasks.

## 3 BACKGROUND

**Lewis Referential Game.** The Lewis referential game is played by two agents, a speaker denoted as S, and a listener denoted as L. The speaker observes a random image from its environment and then sends a descriptive message about the image to the listener. The listener uses this message to guess and reconstruct the image from a set of distractor images. The game's success is evaluated by how well the listener agent guesses the input image. This is a standard game setup for studying emergent language among artificial agents and follows previous works Lazaridou et al. (2018); Li et al. (2020b); Lazaridou & Baroni (2020); Yao et al. (2022). The speaker and listener must communicate and cooperate iteratively to maximize their joint reward.

**Speaker's Message.** At each training step, an input image feature $\mathbf{I}_i \in \mathbb{R}^d$ is randomly selected from the entire set of feature representations for $N$ images, *i.e.*, $\mathcal{D} = \{\mathbf{I}_i\}_{i=1}^N$, where $d$ denotes the dimensionality of the image feature vectors. Similarly, a set of $K$ confounding images (*i.e.*, the distractors) $C_i = \{\mathbf{I}_{ij}\}_{j=1, j\neq i}^K$ is selected from $\mathcal{D}$. The speaker takes the input image feature $\mathbf{I}_i$ and generates a message $\mathbf{M}_i = \langle m_1, m_2, m_i, \ldots, m_T \rangle, m_i \in V$, a sequence of discrete symbols that describes the image, where $T$ is the sequence length limit and $V$ is the message's vocabulary size. The generation process ends when either of the two conditions is satisfied: either the special end-of-sentence symbol [EOS] is generated, or the maximum message length $T_{max}$ is reached.

Initially, at $t = 0$, $m_0 = $ [CLS] and $h_S^0 = \mathbf{I}_i$. At each time step $t > 0$, the generation of the $i$-th speaker message token $m_i^t$ can be described by

$$\mathbf{h}_S^t = \text{GRU}_S\left(m_i^{t-1}, \mathbf{h}_S^{t-1}\right), \tag{1}$$

$$m_i^t = \text{Gumbel-Softmax}\left(\text{MLP}_S(\mathbf{h}_S^t)\right), \tag{2}$$

where the Gumbel-Sofmax trick Jang et al. (2017) is employed to draw samples from categorical distributions of emergent tokens in an end-to-end differentiable way. Here, $\mathbf{h}_S^t$ stands for the hidden state at time step $t$, while $\text{MLP}_S$ denotes the multilayer perception speaker S utilizes to project each hidden state into vectors with dimensionality equal to the vocabulary size of the emergent language.

**Listener's Inference.** The listener agent tries to guess the correct Image $\mathbf{I}_i$ from the set of K confounding images $C_i$, after receiving the generated speaker message $\mathbf{M}_i$. To do so, the listener utilizes a GRU layer to decode the speaker's generated message, *i.e.*,

$$\mathbf{h}_{\mathrm{L}}^0 = \mathrm{GRU}_{\mathrm{L}}\left(m_0, \mathbf{0}\right), \tag{3}$$

$$\mathbf{h}_{\mathrm{L}}^t = \mathrm{GRU}_{\mathrm{L}}\left(m_i^t, \mathbf{h}_{\mathrm{L}}^{t-1}\right), \tag{4}$$

where $\mathbf{h}_{\mathrm{L}}^t$ represents the listener's hidden state at time step $t$.

**Speaker-Listener Optimization.** The listener scores each candidate in the image set, which consists of all distractor images and the correct reference image. Given a message $\mathbf{M}$ and an image $\mathbf{I} \in \{\mathbf{I}_i \cup C_i\}$, the score is defined as

$$\mathrm{score}\left(\mathbf{M}, \mathbf{I}\right) = \left\|\mathbf{h}_{\mathrm{L}}^{|\mathbf{M}|} - \mathrm{MLP}_{\mathrm{L}}(\mathbf{I})\right\|_2^{-2} \tag{5}$$

The likelihood of a selected image $\mathbf{I}_j$ is:

$$p\left(\mathbf{I}_j | \mathbf{M}_i, \mathbf{I}_j, C_i\right) = \left(\frac{\mathrm{e}^{\mathrm{score}(\mathbf{M}, \mathbf{I}_k)}}{\sum\limits_{\mathbf{I}_k \in \mathbf{I}_i \cup C_i} \mathrm{e}^{\mathrm{score}(\mathbf{M}, \mathbf{I}_k)}}\right), \tag{6}$$

where Softmax sampling across scores is used to select an image from the candidate set. The speaker and listener agent's parameters are jointly optimized by maximizing the expected log-likelihood

$$\mathcal{L} = -\mathbb{E}_{\mathbf{M}}\mathbb{E}_{\mathbf{I}_j}\Big[\log p\left(\mathbf{I}_j | \mathbf{M}, \mathbf{I}_i, C_i\right)\Big] \tag{7}$$

After the training of the referential game is complete, the speaker agent model S can be utilized to generate a corpus of emergent language based on the input images. In this work, we are particularly interested in investigating whether the emergent language corpus can be useful as pre-training for cross-modal VL tasks and better understand the emergent language properties that could help in designing more robust and generalizable VL models. We hypothesize that EC tokens, while global in scope, may encapsulate latent information pertaining to specific regions within the image. To evaluate EC corpus transfer on cross-modal matching and reasoning downstream tasks, we generate EC texts for images in Visual Referring Expression (VRE), Visual Question Answering (VQA), and Visual Entailment (VE) benchmarks. In the VRE task, the agent is presented with an image along with a referring expression or caption that identifies a specific object or region within the image, and the agent is required to accurately localize the bounding box corresponding to the referred expression. In VQA, a model is required to reason about an image by understanding its content and subsequently generate articulate responses in natural language to posed questions. Both tasks require the agent to comprehend both modalities effectively and establish a coherent understanding of their mutual interactions for successful task completion. In VE, the agent needs to determine whether a textual hypothesis (*e.g.*, a statement, a question, or a sentence) is true, false, or undetermined based on the visual content depicted in an associated image. For each task, we experiment with pre-training by substituting natural language with emergent text and then fine-tuning on a distinct set of instances with their original natural language annotations.

## 4 EXPERIMENTAL SETUP

### 4.1 VISION LANGUAGUE MODEL

We experiment with a unified vision-language (VL) model that can accommodate various VL tasks. Specifically, we employ OFA Wang et al. (2022a), a Transformer architecture that is suitable for both generation and classification tasks. OFA is pre-trained on multimodal VL datasets, including visual grounding and visual question answering, as well as several unimodal datasets, including image infilling, object detection, and text infilling. These tasks allow the model to develop informative cross-modal representations for relevant input modalities. To better accommodate unified transformer computations, OFA represents images and objects as a sequence of discrete tokens, using a unified vocabulary to represent vision, language, and other modalities. Despite being a sequence-to-sequence

model, OFA is also suitable for classification tasks, such as visual question answering (VQA). During VQA training, a search strategy is used to constrain the generated tokens to a candidate set, which enhances the model's performance Wang et al. (2022a). For the visual grounding task, OFA takes advantage of its generative capabilities by predicting bounding box values directly, without relying on a proposal system. By using a unified approach, OFA can efficiently address various VL tasks with high performance, making it a promising solution for real-world applications. Therefore, we adopt OFA to effectively experiment on various VL tasks.

We evaluate OFA directly on the VRE, VQA, and VE downstream tasks as a baseline without additional pre-training or fine-tuning. Pre-training and fine-tuning OFA on a downstream task dataset generally lead to performance improvements; our primary objective is to determine the minimum achievable accuracy without any additional pre-training. The baselines provide a lower bound for OFA performance. To this end, we assess the performance of the OFA base model on our VRE, VQA, and VE benchmark datasets. Our central goal is to investigate the impact of pre-training a model on the EC corpus and quantify the extent to which it enhances performance on Vision Language tasks. We utilize the officially released OFA base model weights and checkpoints for evaluating the RefCOCO+, VQAv2, and SNLI-VE validation and test sets.

## 4.2 DATASETS

**Visual Referring Expression (VRE).** For the Visual Referring Expression (VRE) task, we leverage the widely adopted RefCOCO datasets Yu et al. (2016) as the standard benchmark. RefCOCO comprises three distinct referring expression datasets (RefCOCO, RefCOCO+, and RefCOCOg) derived from images in the COCO dataset Lin et al. (2014). Each dataset consists of curated referring expressions that describe unique objects or bounding boxes within the images. These expressions were collected by soliciting annotations from human raters who were tasked with identifying objects delineated by bounding boxes in the COCO dataset Kazemzadeh et al. (2014); Yu et al. (2016). RefCOCO+ expressions are strictly appearance-based descriptions (*i.e.*, they do not have location-based captions). For example, "person to the right" is not a valid description for RefCOCO+. Another example is using "the man in the yellow polka-dotted shirt" instead of "the second man from the left". An appearance-based visual grounding dataset is interesting from a computer vision-based perspective since it allows the caption or referred expression to be independent of an observer's view. In detail, the RefCOCO dataset encompasses $142,209$ referring expressions corresponding to $50,000$ objects found in $19,994$ images. RefCOCO+ comprises $141,564$ expressions related to $49,856$ objects spanning across $19,992$ images. Each dataset contains three validation subsets, namely val, testA, and testB. The testA subset corresponds to images featuring people, while the testB subset encompasses images without people. These subsets enable a fine-grained evaluation and analysis of model performance across varying visual contexts. To ensure consistency and comparability with the OFA model Wang et al. (2022a), we adopt their preprocessed version of the RefCOCO datasets. Specifically, each referring expression within the RefCOCO datasets is split into multiple samples, such that each bounding box corresponds to a unique descriptive text.

**Visual Question Answering (VQA).** We conduct experiments on the VQAv2 dataset Goyal et al. (2017). VQAv2 comprises a collection of approximately $1.1$M samples from $200,000$ images accompanied by a 13M natural language answers Goyal et al. (2017). As per OFA, instances for which multiple potential answers exist for a given question-image pair were split into individual samples for each potential answer Wang et al. (2022a). This OFA-adapted version of the VQAv2 dataset includes training, validation, and test sets, with $\sim 1.8$M, $10,402$, and $447,793$ samples, respectively. To further tailor this dataset for EC pre-training experiments, we divide the training set into two parts. The first half is converted into the EC corpus format, wherein each sample's image was processed through the speaker agent to generate a list of emergent tokens that describe the image. These emergent tokens substitute the sample's original natural language answer within the VQA dataset. Meanwhile, the second half of the training set remains unchanged with the original natural language annotations and is used for VQA fine-tuning.

**Visual Entailment (VE).** In VE, an image $\mathbf{I}$ serves as a premise along with a hypothesis text $\mathbf{H}_{\text{text}}$. The objective is for the model to determine whether the hypothesis can be inferred from the image (*i.e.*, if the image entails the hypothesis) Xie et al. (2019). The model's output is classified as an *entailment* (if there is sufficient evidence in $\mathbf{I}$ to conclude that $\mathbf{H}_{\text{text}}$ is true), a *contradiction* (if there is

enough evidence in $\mathbf{I}$ to conclude that $\mathbf{H}_{\text{text}}$ is false), or *neutral* (if there is insufficient evidence in $\mathbf{I}$ to make a conclusion about $\mathbf{H}_{\text{text}}$). VE shares similarities with VQA as both tasks require the model to reason and make logical deductions based on an image. However, VE is more complicated than VQA, in the sense that choosing whether a text entails an image requires complex fine-grained reasoning beyond answering a question. To evaluate the performance of EC pre-training on VE, we conduct experiments on the SNLI-VE dataset Xie et al. (2019; 2018). This dataset comprises $529, 527$ training samples, $17, 858$ validation samples, and $17, 901$ test samples generated from $29, 783$ distinct images. To assess the impact of EC pre-training under varying training sizes, we further partition the training set into full-size, $50, 000$, and $10, 000$ samples. Experimental results for these subsets compare the effectiveness of EC pre-training in enhancing VE performance under different training conditions.

## 4.3 EC PRE-TRAINING

**Visual Referring Expression (VRE).**   We pre-train the OFA base model for VRE on the RefCOCO train set. We use the EC-generated text from the referential game speaker agent as captions for pre-training. To obtain the EC text, we employ a ResNet-18 model He et al. (2016) and extract 512-dimensional image features from each image in the RefCOCO training set. These image features are then passed to a speaker agent that generates a set of Emergent Communication (EC) tokens describing the image. Finally, we replace the original natural language RefCOCO caption that describes a bounding box with the EC-generated tokens. As the EC tokens encompass information about the image, we hypothesize that they also contain some information about the specific bounding box, thus offering potential pre-training benefits. The resulting RefCOCO dataset, post-processed with EC tokens as captions, serves as the basis for pre-training the OFA base VRE model.

**Visual Question Answering (VQA).**   In addition to the grounding task, we explore EC pre-training for the VQA reasoning task. Reasoning tasks not only require a connection between visual and linguistic information but also logical deductions. VQA requires interpreting visual input within the context of a question problem posed in natural language to produce a language-based answer. The VQAv2 dataset Goyal et al. (2017) provides English language annotations for each question-answer pair, enabling models to learn directly from the structure and vocabulary imposed by English. Here, we investigate the impact of pre-training the OFA model on EC text concerning the model's ability to acquire the necessary prerequisites for reasoning tasks, even without precise natural language information. We employ a ResNet-18 model to extract 512-dimensional image features from each image in the VQAv2 training set. These image features are then fed into the speaker to generate EC tokens containing sufficient information to describe the provided image. We perform pre-training on the VQA task. Here, the EC text should contain sufficient information to describe the provided image. The pre-training process trains the model to generate EC answers, which could prime the model to associate certain characteristics of the input image with certain outputs —an essential aspect of VQA.

### 4.3.1 NL FINE-TUNING

**Visual Referring Expression (VRE).**   We fine-tune the best-performing EC-pretrained model checkpoint on RefCOCO+. By using different visual grounding datasets in pre-training and fine-tuning stages (RefCOCO for pre-training and RefCOCO+ for fine-tuning), we ensure that information does not overlap between the two phases. Subsequently, we evaluate the performance of the finetuned model on all RefCOCO+ test sets, including val, testA, and testB. For NL fine-tuning, we fine-tune the OFA base model on the VRE task, employing natural language captions from the RefCOCO+ dataset. We consider this as an upper-bound model for comparison against the EC pretraining experiment. Since the OFA model has already been trained on a diverse and extensive range of multimodal datasets, including the RefCOCO visual grounding dataset, additional pre-training on RefCOCO would not yield significant benefits. Following fine-tuning, we evaluate the finetuned model performance on all RefCOCO+ validation and test sets.

**Visual Question Answering (VQA).**   We fine-tune the best-performing EC-pre-trained checkpoint model on the remaining unaltered VQAv2 dataset with natural language question-answer pairs and evaluate on the VQAv2 validation and test sets. Similarly to VRE, for direct NL fine-tuning, we fine-tune the OFA base model on the natural language split of the VQAv2 dataset and consider this as an upper bound for comparison against EC pretraining. Subsequently, we evaluate the fine-tuned model on the VQAv2 validation and test-dev sets.

Table 1: VRE and VQA accuracy. VRE results include three RefCOCO evaluation sets: testA, testB, and val, while VQA results are reported on the VQAv2 validation and test-dev sets. The three columns in each table represent our baseline and pre-training approaches: base, +EC Pretraining, and +NL Pretraining, which correspond to directly evaluating OFA model without fine-tuning, EC pre-training followed by NL finetuning and NL Pretraining followed by NL fine-tuning, respectively.

<table>
<tr><td colspan="4">(a) Visual Referring Expression (VRE)</td></tr>
<tr><td></td><td>Base</td><td>+EC</td><td>+NL</td></tr>
<tr><td>val</td><td>29.81</td><td>62.17</td><td>81.91</td></tr>
<tr><td>testA</td><td>31.49</td><td>67.15</td><td>86.60</td></tr>
<tr><td>testB</td><td>27.53</td><td>51.16</td><td>73.49</td></tr>
</table>

<table>
<tr><td colspan="4">(b) Visual Question Answering (VQA)</td></tr>
<tr><td></td><td>Base</td><td>+EC</td><td>+NL</td></tr>
<tr><td>val</td><td>66.30</td><td>56.52</td><td>74.65</td></tr>
<tr><td>test-dev</td><td>70.01</td><td>49.28</td><td>73.40</td></tr>
</table>

**Visual Entailment (VE).** To assess the potential of emergent communication (EC) models for transferring knowledge to new vision language tasks, we employ the EC pre-trained model obtained from VRE as pre-training for the VE task. We fine-tune the best EC pre-trained model checkpoint on the SNLI-VE training set and evaluate on the SNLI-VE test and dev validation sets. To further understand the impact of EC under varying sampling conditions, we conduct fine-tuning on randomly sampled sets with different sizes, *e.g.*, $10,000$ and $50,000$ samples. For the NL experiment, we fine-tune the OFA base model on SNLI-VE. To assess the impact of training set sample size on model performance, we similarly perform fine-tuning with different training set sample sizes and evaluate our models on the SNLI-VE validation and test sets.

## 5 EXPERIMENTAL RESULTS

**Visual Referring Expression (VRE).** Table 1a presents results for the VRE task. Evaluation includes the baseline (**Base**), pre-training using EC captions (**+EC**), and pretraining using NL captions (**+NL**), respectively. Both EC-pretrained and NL-pretrained OFA models were finetuned on the RefCOCO+ natural language dataset. Pre-training on EC captions significantly improves the VRE task's performance compared to the baseline model. This highlights the valuable transfer benefits and shared knowledge that can be extracted from Emergent communication tokens when utilized for pre-training in vision-language tasks. As expected, NL (Natural Language) pretraining significantly outperforms both the baseline and EC pretraining methods due to the inherent structural and grounding advantages of natural language. However, it is noteworthy that pretraining the VRE model on the EC corpus achieves a substantial two-thirds of the performance achieved through pretraining with natural language. This underscores the potential utility of EC tokens in vision-language pretraining, especially in settings where natural language data is scarce. The EC referential game and the generation of EC tokens by the speaker agent can be performed on unlabeled images in the wild, effectively producing descriptive captions for these images. Consequently, EC tokens contain valuable semantic information about the images, which can prove advantageous for vision-language pretraining tasks. These findings emphasize the potential of leveraging EC pre-training for vision-language tasks. We also provide qualitative examples of the VRE task. Figure 2 presents examples from the RefCOCO dataset, where each image is associated with the corresponding emergent and natural language referring expression. Additional examples and discussion can be found in the supplementary.

**Visual Question Answering (VQA).** Table 1b presents results for the VQA task. During EC pre-training, target answers are replaced with EC text, offering an opportunity to learn additional visual and structural information without requiring full comprehension of the context provided by the natural language question. Results show a decrease in performance when utilizing EC pretraining for the VQA task compared to the OFA pretrained checkpoint (base), indicating that further usage of emergent text after natural language pretraining has already been performed may lead to a degradation in the already learned representations. In contrast, experiments with models pretrained from scratch (Appendix C) exhibit the opposite effect. In this case, EC pretraining improved downstream VQA performance compared to no pretraining, suggesting that when no natural language pretraining data is available, EC pretraining can still be beneficial for improving VQA performance.

**Visual Entailment (VE).** In Figure 3, we present experimental results for fine-tuning the EC-VRE pre-trained model on visual entailment. Our results demonstrate that pre-training a vision language model on EC tokens leads to a significant increase in accuracy performance for the VE task across

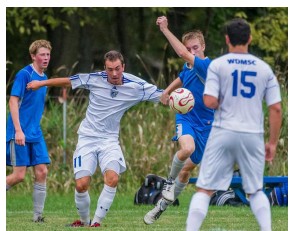 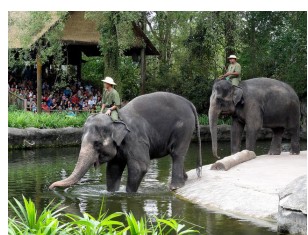 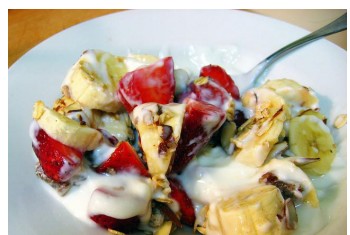

(a) White uniform left.
[2358 3708 3638 2557 3563 3731 1270 1357 2251 1037 829 3958 3352 629 0]

(b) Right elephant.
[1215 318 3702 1583 608 3672 787 3647 1451 1508 2777 380 1733 2522 0]

(c) Top left banana piece.
[1692 3465 828 2906 2108 3307 1509 3767 101 1010 2694 3134 1508 3823 0]

Figure 2: Qualitative examples on the VRE task. RefCOCO images and their corresponding emergent and natural language referring expressions.

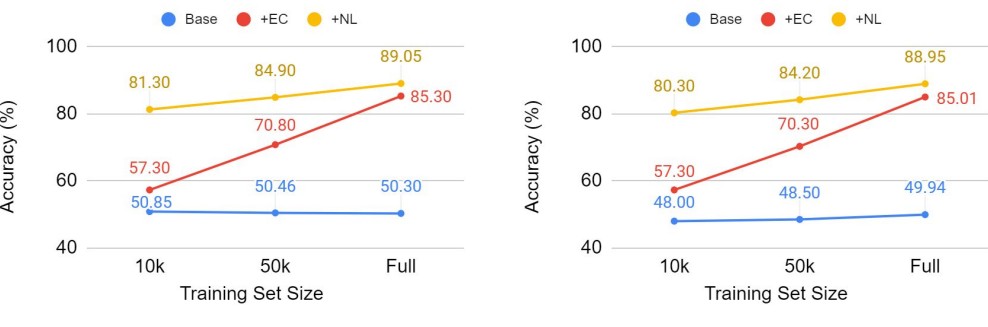

(a) SNLI-VE validation set

(b) SNLI-VE test set

Figure 3: Visual Entailment (VE) accuracy. Ablation with varying training set sizes and different pre-training/fine-tuning methods: **Base**, **+EC Pretraining**, and **+NL Pretraining**, corresponding to training from scratch with no pre-training, EC pre-training followed by natural language (NL) fine-tuning, and NL pre-training followed by NL fine-tuning, respectively.

various training sizes and validation sets when compared with the baseline model. These results provide compelling evidence of the utility of EC tokens for transferring knowledge between VL tasks.

When comparing EC pre-training with the natural language fine-tuning of OFA, it is expected that the latter would achieve better performance. However, it is surprising to observe that EC pre-training can yield comparable performance in certain settings. For instance, when fine-tuned on the full training data, EC achieves an accuracy of $85.01\%$ on the test set, whereas NL achieves $88.95\%$. The relatively small performance gap between EC-VRE pre-training and natural language fine-tuning highlights the transferability benefits of EC pre-training and opens up future directions in utilizing emergent language for complex vision language tasks and combining the strengths of emergent language with the structure and complexities of natural language for better and more efficient VL modeling.

## 6 CONCLUSION

Whereas several previous VLM works focus on building better-performing models by improving image and text encoding mechanisms, designing performant architectures, and devising effective pre-training tasks, our work introduces a paradigm shift by considering improving vision language learning through Emergent Communication (EC) pre-training. We explore the potential benefits of EC pre-training for vision language (VL) downstream tasks, specifically visual referring expression, visual question answering, and visual entailment. We observe substantial improvements in learning and generalization on certain tasks after fine-tuning with a VL model that was pre-trained on VL datasets with EC tokens, generated from a speaker agent trained during an EC referential game. Our findings highlight that EC is promising in acting as an inductive bias for Vision Language learning and can yield substantial transfer benefits for low-resource tasks and data-constrained settings.

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
