## A    APPENDIX

## B    IMPLEMENTATION DETAILS

The EC speakers used to generate the EC datasets are directly trained on the COCO image features from the codebase[1] of Yao et al. (2022) for 2000 epochs. Training is performed on a P100 GPU, and the sequence length limit is set to 15. Generating EC sequences of length 15, the speakers draw from a vocabulary size of 4035 tokens. In the Visual Referring Expression (VRE) experiments, we adopt the codebase of OFA[2] Wang et al. (2022a) and mostly follow their default setup. Initially, we pre-train the OFA model on the RefCOCO training set with EC annotations via continuous pretraining Wang et al. (2022a). The pretraining data was prepared to align with OFA's structure, as outlined on their GitHub page. The pretraining process consists of 17 epochs and 492,000 updates. Subsequently, we fine-tune the pre-trained model for 10 epochs and 18,500 updates. Pretraining was conducted on a single NVIDIA A100 GPU for 2 days, while the finetuning phase required 2 NVIDIA A100 GPUs and took 2 days to complete. For the Visual Question Answering (VQA) task, we again adopt the OFA codebase and follow the default setup provided in both the continuous pretraining and VQA finetuning and evaluation scripts. Continuous pretraining was executed on a single NVIDIA V100 GPU for 4 days, encompassing 960,000 updates, which corresponded to approximately 4 to 5 epochs. Finetuning for 5 epochs required around 90 hours on 2 P100 GPUs. In the Visual Entailment (VE) task, we utilize the pre-trained model that was obtained in the VRE task. We finetune this model on the SNLI-VE dataset Xie et al. (2019) for 5 epochs and 20,500 updates. The fine-tuning process was distributed across 4 NVIDIA A40 GPU workers and took approximately 15 hours to complete. For the rest of the training process, we largely adhered to the OFA setup.

## C    ADDITIONAL EXPERIMENTS

### C.1    PRETRAINING ON MULTIPLE TASKS

To further test the contribution of EC text, we perform experiments with alternative settings. In these experiments, we combine the original natural language versions of the datasets utilized in the previous experiments into a unified NL pretraining dataset, while also amalgamating their EC counterparts into a consolidated EC dataset. Both of these datasets encompass samples from Visual Referring Expression (RefCOCO), Image Captioning (MSCOCO), and Visual Question Answering (VQAv2, 1st half). In contrast to the previous experiment, where the OFA checkpoint was pretrained on an abundance of natural language text, all models in this alternative setup begin pretraining from scratch. In this paradigm, we compare OFA models produced from each pretraining dataset, *i.e.*, a model that is exclusively pretrained on NL samples with its counterpart pretrained solely on EC samples. After pretraining, the best-performing checkpoints for each model variant (EC and NL) are finetuned independently on the three downstream vision-language tasks (VRE, VQA, and VE). The baseline is an OFA model with randomly initialized weights trained from scratch on each task, without any pretraining. These experiments aim to elucidate the added value of EC text in comparison to models trained without pretraining or those pretrained solely with NL data, thus offering a comprehensive understanding of the role of emergent language in enhancing vision-language tasks.

The results presented in Table 2 highlight the potential of EC pretraining as a foundational or initial model for training on diverse multimodal tasks. In the context of VRE, EC pretraining surpasses a baseline model trained directly on visual grounding tasks without additional pretraining, achieving an impressive accuracy gain of over $108\%$. Similarly, in VQA, we observe a noteworthy $11.5\%$ performance improvement on the test-dev set. We note, however, that there is a slight decline in performance between the baseline model and the EC pre-trained model in the case of VE. This discrepancy can be attributed to the categorical nature of VE, where the model aims to categorize hypotheses into one of three classes: Entailment, Contradiction, or Neutral. As anticipated, NL pretraining exhibited superior performance across all tasks. It is important to clarify that our objective is not to establish EC as a replacement for NL pretraining, given that natural language inherently possesses a higher degree of structure and organization compared to EC language. Instead, this

---

[1] https://github.com/ysymyth/ec-nl/tree/master/ec-game
[2] https://github.com/OFA-Sys/OFA

Table 2: **Base (No Pretraining)** - Fine-tuning OFA on natural language RefCOCO+ (VRE), VQAv2 (VQA), and SNLI-VE (VE) train sets without pretraining. **+EC Pretraining** - Fine-tuning the EC pre-trained model and **+NL Pretraining** - Fine-tuning the NL pre-trained model, both fine-tuned on natural language RefCOCO+ (VRE), VQAv2 (VQA), and SNLI-VE (VE) train sets.

|     |          | **Base** | **+EC** | **+NL** |
|-----|----------|------|-------|-------|
| **VRE** | **val**  | 10.03 | 23.77 | 40.15 |
|     | **testA**   | 13.88 | 28.89 | 45.90 |
|     | **testB**   | 9.72  | 18.84 | 31.34 |
| **VQA** | **val**      | 49.33 | 50.61 | 56.66 |
|     | **test-dev** | 40.80 | 45.49 | 51.26 |
| **VE**  | **dev**      | 78.31 | 77.00 | 80.09 |
|     | **test**     | 78.60 | 77.07 | 80.21 |

work investigates whether emergent language offers any structural and semantic advantages that can enhance vision-language understanding, especially in scenarios characterized by a scarcity of natural language data. Overall, the findings indicate that EC pretraining can be valuable and could offer substantial benefits in certain vision-language tasks (*e.g.*, VRE) but may not be universally superior to NL pretraining, which remains the upper bound in terms of performance across these tasks. Future research in this area can build upon these results to further explore the potential of Emergent Communication (EC) pretraining in vision-language models, *e.g.*, design hybrid pretraining methods that combine EC and NL pretraining, and explore improvements in EC fine-tuning strategies.

## D    UNIGRAM DISTRIBUTIONS

We conduct a comparative analysis of the unigram distributions between the corpus generated from natural language captions of the entire RefCOCO training dataset and the corpora of EC text generated from EC speakers producing text sequences of lengths 5, 15, and 25. The EC corpora were generated from the first 10,000 samples of the RefCOCO training set. To obtain the unigram distribution for both natural language and EC text, we utilize the nltk word tokenizer[3] to tokenize each piece of text, whether it was a natural language caption or a string of EC tokens. The unique tokens were then counted and sorted to generate the respective unigram distributions for each corpus.

Figure 4 illustrates the unigram distributions derived from the RefCOCO dataset, showcasing the contrasting characteristics of the natural language and EC text corpora. Additionally, Figure 5 presents the unigram distributions based on the NL-based VQAv2 training set questions and answers, along with the EC distribution derived from the first 10,000 samples of the EC-based VQAv2 training set. The analysis reveals that the natural language corpus encompasses a wide variety of tokens in comparison to the EC corpora. Moreover, the EC text employs a greater number of unique tokens as the sequence length becomes shorter. This observation suggests that the EC speakers are adept at utilizing positional information to reuse tokens while maintaining their descriptive power. This is further confirmed by observing that fewer tokens are used relatively more frequently with a sequence length of 25 as compared to sequences of length 15.

## E    INVESTIGATING THE IMPORTANCE OF STRUCTURE IN EC LANGUAGE

As shown in 3, the model pre-trained on the original EC language improves downstream performance more than the model pretrained on shuffled EC and random EC language in most of the test cases. This points to the significance of structure and semantic grouding of EC in vision language pretraining. In order to gain insights into the generalization capabilities and support of vision language pretraining by EC language, we conduct an ablation analysis focusing on the significance of structure and semantic grounding. For this purpose, we pre-train a cross-modal task agnostic OFA model on three types of EC language: 1) $EC_{orig}$: the original EC language generated by the referential game speaker, 2) $EC_{reordered}$: shuffled or reordered EC language (to assess the impact of structure), and 3)

---

[3] https://www.nltk.org/api/nltk.tokenize.html

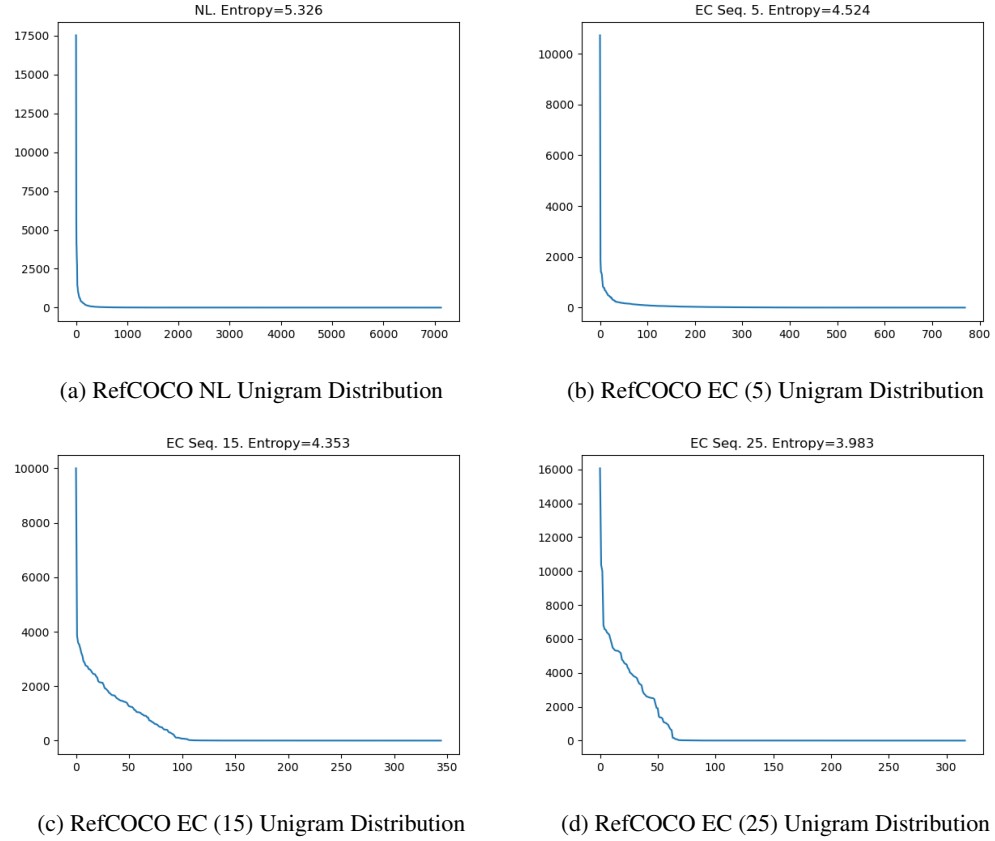

Figure 4: Unigram distributions and entropies for the RefCOCO dataset on NL and EC sequences.

Table 3: Ablation analysis of the significance of structure and semantic understanding in the EC language. $EC_{orig}$, $EC_{reordered}$, $EC_{random}$ represents original EC language generated by speaker, perturbed EC language by reshuffling, and random EC tokens, respectively.

| | Base | $EC_{orig}$ | $EC_{reordered}$ | $EC_{random}$ |
|---|---|---|---|---|
| **val** | 29.81 | 42.50 | 42.10 | 42.10 |
| **testA** | 31.49 | 47.68 | 46.91 | 49.16 |
| **testB** | 27.53 | 34.49 | 33.63 | 33.34 |

$EC_{random}$: random EC language (to assess the influence of semantic grounding). Due to computational constraints, and for all variations, we use 30k samples of the RefCOCO pretraining data instead of the original 120k used in the main paper. The 30k samples from the initial EC annotations are denoted as $EC_{orig}$. We then shuffle the EC annotations to generate the pre-train data $EC_{reordered}$. Finally, we replace the original annotations with a sequence of random numbers of the same sequence length and use this as the $EC_{random}$ pre-train data. After pretraining is complete on each of these EC language variants, we finetune each resulting pre-trained model on 10k RefCOCO samples. As shown in Table 3, the model pre-trained on the original EC language exhibited higher performance on most evaluation splits compared to the models pre-trained on shuffled EC language and random EC language. These findings underscore the importance of both structure and semantic grounding in EC language for effective vision language pretraining.

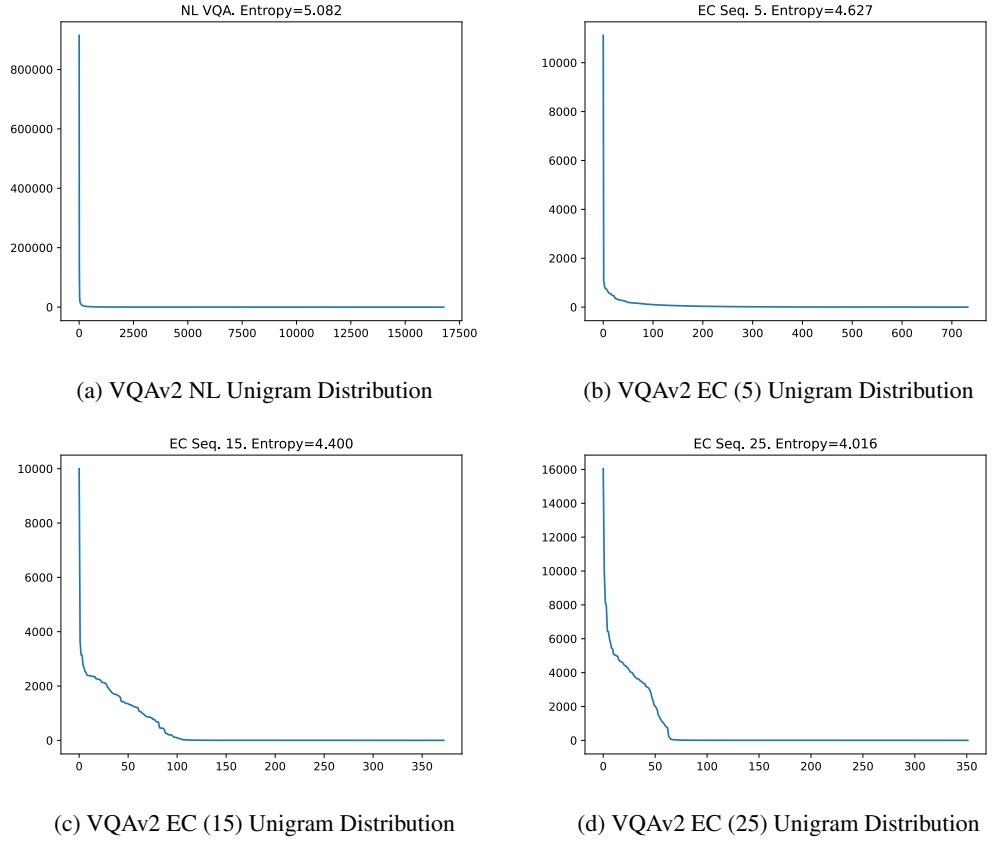

(a) VQAv2 NL Unigram Distribution

(b) VQAv2 EC (5) Unigram Distribution

(c) VQAv2 EC (15) Unigram Distribution

(d) VQAv2 EC (25) Unigram Distribution

Figure 5: Unigram distributions and entropies for the VQAv2 dataset on Natural Language and EC sequences.

## F    QUALITATIVE EXAMPLES

We conduct an in-depth qualitative analysis to uncover potential patterns in the generated Emergent Communication (EC) text. For this analysis, we train three EC speakers over 1000 epochs, utilizing a vocabulary size of 4035 and sequence lengths of 5, 15, and 25. The speakers are trained using COCO features from the EC game introduced by Yao et al. (2022). To generate EC text, we pass the first 1000 unique images from the refCOCO training dataset through each speaker. To focus our evaluations, we track the positions of EC text n-grams within the generated sequences. If more than 5 images produce the same n-gram in the same position of the text sequence, we group those images together for manual observation. Specifically, we examine bigrams, trigrams, and 4-grams for EC sequences of length 5, 15, and 25, respectively.

Figure 6 shows a few groupings found in text sequences of length 5. Notably, token 2430 is strongly associated with broccoli, while token 222 is frequently utilized to describe food as a broader category earlier in the sequence. In Figure 7, which presents examples with a sequence length of 15, we observe that token 3293 exhibits a strong association with zebras. Figure 8 further illustrates how the same tokens, when placed in different positions, convey similar yet more refined meanings. For instance, token 309 corresponds to vehicles, but its count and position within the sequence determine whether it refers to a truck or a motorbike. Furthermore, we observe that token 3355 appears in numerous sequences, suggesting its potential role in providing spacing to indicate structural meaning.

Additionally, Figure 9 demonstrates a relationship between n-gram structure and overall sequence structures. Notice that the 1599 1599 bigram is consistent throughout these sequences, except one is in position 0 (1599 1599 x x x) and the other in position 1 (x 1599 1599 x x). This seems to indicate that the n-gram structure is more important than the explicit place in the sentence, or that there is a

larger structure that is not captured in this qualitative analysis. Some images in Fig. 9b are also in Fig. 9a indicating trigram (1599 1599 1599 x x). This could be an indicator of a hierarchical structure in which a token can be repeated to increase the specificity of a particular feature. Additionally, we observe instances where the same bigram appears in a different position of the sequence but describes the same image. For example, image 50 in 9 can be seen in bigrams (1599 1599 x x x) and (x 1599 1599 x x), indicating that image 50 has the trigram (1599 1599 1599 x x). Similar to natural language, due to context and semantic relationships between the words involved, a subsequence of words that represents a fixed phrase with a specific meaning can describe the same image or convey the same visual aspects, even if the position of that subsequence within the sentence is shifted.

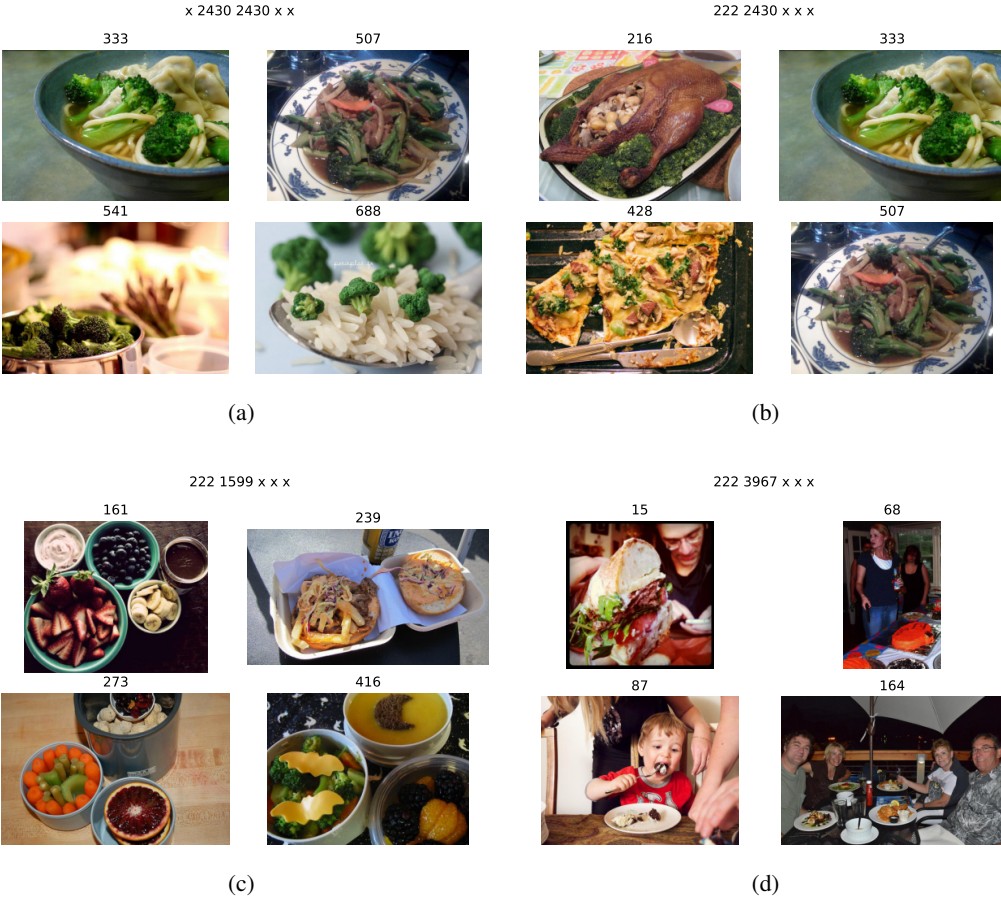

Figure 6: EC sequences of length 25, with EC tokens relating to food. (a) The repeated 2430 token is related to broccoli. (b) Token 222 is associated with the wider category of food. (c) Changing tokens after 222 changes what kind of food is described. (d) Bigram 222 3967 is still associated with food, but also people near to or eating it.

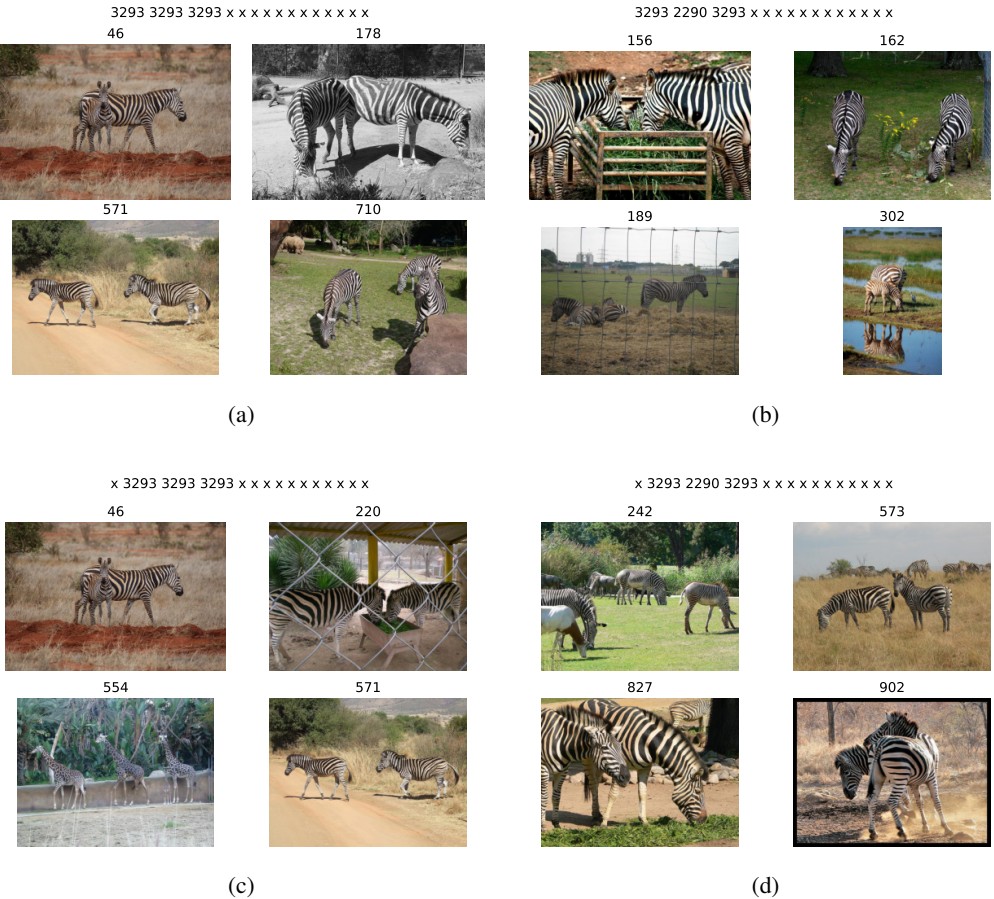

Figure 7: EC sequences of length 5. Tokens denoting patterned animals, especially zebras. (a) Token 3293 strongly indicates zebras. (b) A different token in the trigram may indicate something that distinguishes these zebras from other images. (c) The same trigram as (a) but in a different position. Giraffes are also lightly associated with these tokens. (d) The trigram of (b) in position 1 is still associated with zebras.

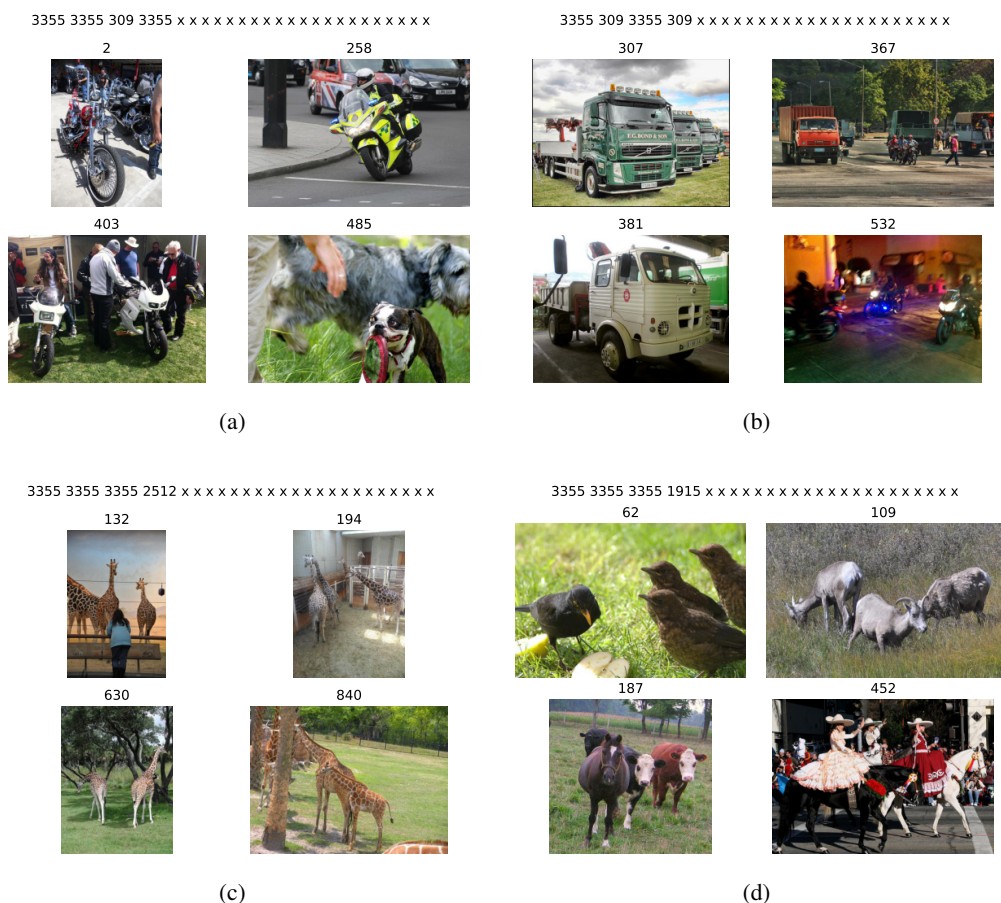

Figure 8: EC sequences of length 25. (a) Token 309 is associated with vehicles. When in position 2, it is associated with motorbikes. (b) Token 309 in different positions still indicates vehicles, but this time denotes trucks. (c) Perhaps token 2512 denotes giraffes here and (d) token 1915 denotes a wider class of animals. Note that token 3355 has consistently been used in (a)-(d) indicating that it could be a structural filler or other indicator, for example, it could be a numeric indicator conveying whether there are few or many instances of an object in the image.

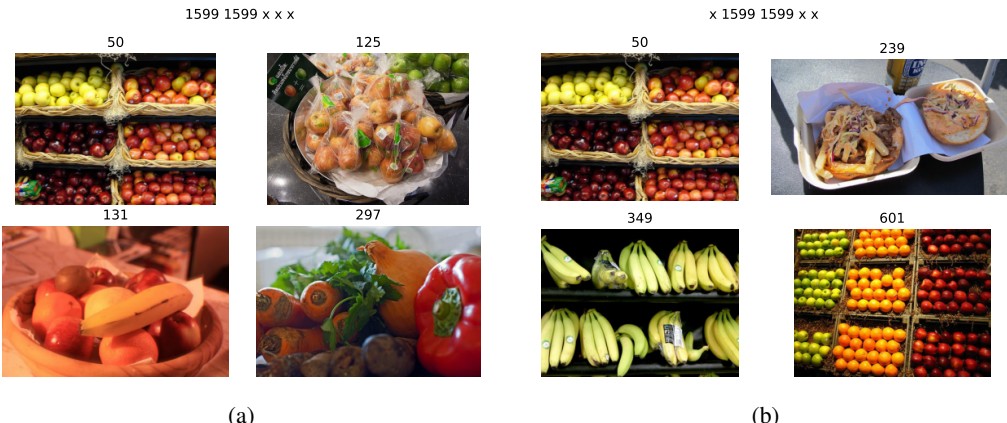

Figure 9: Images grouped with what seems to be a yellow, red, and green color palette, especially food. (a) Bigram 1599 1599 in position 0 shows fruit and vegetables with these colors. b) When in position 1, this bigram denotes a smaller set of images with a similar color scheme.

# G    DISCUSSION

**Limitations.**    It is important to acknowledge that our exploration of the benefits of EC pre-training is not exhaustive of the wide range and variety of Vision Language tasks. Although we specifically focused on VRE, VQA, and Visual Entailment, our findings revealed significant improvements in learning and task generalization. Nonetheless, our proof-of-concept sets the stage for future investigations into the advantages of EC pre-training across additional Vision Language tasks, such as image-text retrieval (ITR), image-text matching (ITM), and Grounded Captioning (GC), among others. One potential limitation of our approach is its current inability to generate EC text for various modalities and scenarios beyond image description. While the explored referential game effectively trains the EC speaker to describe images, the structure and semantics of the generated language may be tailored predominantly toward literal image descriptions. Consequently, the substitution of natural language (NL) text with EC text for certain datasets or tasks may pose challenges. This limitation prompts further exploration into expanding the applicability of EC pre-training beyond image description, ensuring its compatibility with a broader array of modalities and capturing diverse linguistic aspects. In a broader context, our work underscores the potential of harnessing the strength of EC tokens, which are more accessible in data-constrained settings, in conjunction with structured and complex natural language corpora. This fusion enables improved and promising vision language learning, yet it also highlights the need for continued research into optimizing the integration of EC tokens with more complex and comprehensive natural language resources.

**Broader Impact.**    Our work has several potential broader impacts. Firstly, the possibility of integrating emergent language pre-training from Emergent Communication (EC) into Vision Language (VL) models, paves the way for the development of more robust and generalizable VL models. This could have a positive impact on various applications, such as image-text retrieval, visual search, visual question answering, and image captioning, in addition to important implications for enabling VL models to perform effectively in real-world settings where representative data is limited, thus enhancing their practical utility. Secondly, our research contributes to advancing communication between humans and machines. By investigating how agents learn to communicate in EC games and establishing connections with vision language systems, we gain deeper insights into the cognitive and computational mechanisms that underlie effective communication. This understanding can fuel the development of more efficient and intuitive communication systems, benefiting both humans and machines in various domains. Thirdly, our work contributes to the development of new artificial intelligence (AI) technologies. Unraveling the ways in which EC can enhance VL models can lay the foundation for the creation of AI systems that possess enhanced learning and comprehension capabilities. This, in turn, contributes to the evolution of AI technologies that better understand and interact with the world around them. We anticipate minimal foreseeable negative impact associated with our work, as the nature of emergent language differs from natural languages. For instance, the emergence of bias in data is less likely as the emergent language is generated by AI agents, in contrast to natural languages that can exhibit inherent biases and prejudices. Consequently, our research primarily focuses on the positive implications that emergent communication offers for advancing AI technologies and facilitating effective human-machine communication.