# OpenReview forum: "Emergent Corpus Pretraining Benefits Vision Language Modeling"
_ICLR.cc/2024/Conference — ICLR 2024 Conference Withdrawn Submission_

### Official Review · Reviewer_DU5f · 2023-10-14

**Soundness:** 1 poor
**Presentation:** 2 fair
**Contribution:** 1 poor
**Rating:** 3
**Confidence:** 4

**Summary:**

The paper explores the problem of generalization in vision-language modeling. To enhance the generalizability of vision-language models, the authors propose to use emergent communication (EC) between a listener and a speaker agent. Experiments showcase several potential benefits of EC for the visual entailment, visual question answering, and visual referring expression tasks.

**Strengths:**

The paper includes the following strengths:

- The proposed method is intuitive and interesting.

- The methodology is well-written and understandable with accurate notations and coherent explanations.

- The experiment section lucidly states the details regarding the settings and results.

**Weaknesses:**

The paper contains some minor but serious weaknesses:

- The motivation for EC is not convincing. For example, in the introduction, the author declares that EC is a promising approach without providing any evidence / explaining why this is promising. Moreover, even though self-supervised learning (SSL) is a well-known solution to tackle the limit of labeled data, the introduction lacks the the discussion towards SSL.

- The intuition of emergent language is not evident. Why do some unintelligible tokens, e.g. in Figure 2, can benefit the vision-language models, which primarily work with natural language? The paper does not provide an intuitive discussion towards this aspect.

- The experiments are also not comprehensively conducted. There is not an ablation study to investigate the effect of each component and also an analysis for better understanding the EC framework, e.g. why the authors choose OFA as the base model for EC?

**Questions:**

- In the intuitive sense, why does emergent language can benefit vision-language modeling, which is mainly about natural language?

- Why do you choose OFA as the base model? Does EC perform effectively with other models, such as UniVL, CLIP, etc.?

---

> ### Author Response · Authors · 2023-11-22
> **Thank you for your constructive review**
>
> Thank you for finding our method “intuitive and interesting”. We have addressed all of your questions below. For a more comprehensive view of our work, please also refer to our response to other reviewers.
>
> **The motivation for EC is not convincing. For example, in the introduction, the author declares that EC is a promising approach without providing any evidence / explaining why this is promising.**
>
> We appreciate your feedback regarding the motivation for emergent communication (EC) in our work. Recognizing the importance of addressing this concern, we have thoroughly revisited the introduction to provide a more comprehensive explanation of why EC pretraining is a promising approach to address certain limitations in Vision-Language Models (VLMs). Please also see our response to reviewer 2A8R.
>
> - **Natural language substitution in low resource settings:** Emergent communication (EC) shares several properties with natural language (NL), as highlighted by Li et al. (2020) and Yao et al. (2020), encompassing aspects such as compositionality, learning, structure, and symbolic representation. This similarity motivates us to explore EC pretraining since if emergent communication proves effective for VLM-PTMs, it could be a valuable tool for training models when large labeled parallel vision-language datasets are scarce or expensive to obtain.
>
> - **The structural and semantic benefit of EC:** The Lewis referential game, a learned emergent communication (EC) process, unfolds in a multimodal environment, tightly integrating vision and language. In this game, a speaker agent processes an image and generates an EC message to describe it. Consequently, the emergent messages encapsulate both visual and linguistic stimuli, providing a valuable learning signal that can benefit vision-language modeling and enhance the model's ability to understand and generate meaningful associations between visual and linguistic elements.
>
> - **Past successes in Language modeling:**  Drawing on past successes in language modeling, Li et al. (2020) demonstrate the advantages of emergent communication in pretraining for natural language tasks. This ICLR 2022 spotlight work showcased non-trivial gains in language modeling perplexity through pretraining on a corpus of emergent language. Building upon this foundation, our research extends the applicability of emergent language to vision-language learning, emphasizing its broader utility and effectiveness in multimodal contexts
>
>
>
>
> **Moreover, even though self-supervised learning (SSL) is a well-known solution to tackle the limit of labeled data, the introduction lacks a discussion about SSL.**
>
> We have improved the introduction section of our paper and added a discussion on the role of SSL in tackling labeled data and how this leads to the motivation for our research. We note that our initial paper version covered a few key SSL papers in the related work section (see Pre-training for Vision Language Transfer Learning).
>
>
> **References**
> - Yao et al. (2022) Linking Emergent and Natural Languages via Corpus Transfer. ICLR 2022 Spotlight
> - Li et al. (2020) Emergent Communication Pretraining for Few-Shot Machine Translation. COLING 2020.

---

> > ### Author Response · Authors · 2023-11-22
> > **Continued response**
> >
> > **The intuition of emergent language is not evident. Why do some unintelligible tokens, e.g. in Figure 2, can benefit the vision-language models, which primarily work with natural language? The paper does not provide an intuitive discussion towards this aspect.**
> >
> > The emergent tokens encompass structures that carry learning signals capable of enhancing vision-language modeling. Similar to natural language, these tokens possess the potential to contribute to machine intelligence. As previously highlighted in our response to the first concern, emergent tokens share numerous similarities with natural language, particularly in terms of pre-training benefits. *We would appreciate it if the reviewer could mention specifically why they think the tokens are unintelligible.* In our supplementary section F, we conduct an extensive analysis to uncover patterns within the generated EC tokens, offering insights into their structural and semantic characteristics. In this analysis, we reveal patterns associating certain tokens with categories such as food, animals, and vehicles. Additionally, we observe instances where tokens, when placed in different positions within a sequence, convey similar yet more refined meanings. For instance, Figure 9 illustrates that token 309 corresponds to vehicles, with its count and position influencing whether it refers to a truck or a motorbike. Furthermore, token 3355 appears consistently across various sequences, suggesting its potential role in providing spacing to indicate structural meaning. These observed intrinsic structural properties of EC tokens provide a basis for their positive impact on vision-language learning. We believe that the patterns identified in our supplementary analysis contribute to the intuitive understanding of how emergent tokens can carry meaningful information, aligning with the goals of vision-language modeling. We hope this additional clarification addresses your concerns, and we welcome further feedback on this aspect. Thank you for your valuable insights.
> >
> > **The experiments are also not comprehensively conducted. There is not an ablation study to investigate the effect of each component and also an analysis for better understanding the EC framework.**
> >
> > We would like to highlight that we have conducted several ablation studies and analyses to thoroughly investigate the components and framework of emergent communication (EC). These studies are detailed in different sections of our supplementary material:
> > - Section D: In this section, we perform a comparative analysis of the token unigram distributions between the corpus generated from natural language captions and the corpus of EC text. Our findings indicate that EC speakers effectively utilize positional information to reuse tokens while preserving their descriptive power. This analysis provides insights into the efficiency and characteristics of token usage in EC-generated text.
> > - Section E: To gain insights into the generalization capabilities of EC pretraining for vision-language tasks, we conduct an ablation analysis in Section E. This analysis focuses on the significance of structure and semantic grounding. We compare models pre-trained on the original EC language with those pre-trained on shuffled EC and random EC language. The results demonstrate that the model pre-trained on the original EC language consistently outperforms its counterparts in downstream tasks, highlighting the importance of preserving structure and semantics in EC pretraining.
> > - Section C.1: Within this section, we explore different experimental settings with a stronger baseline model. All models in these settings start from scratch and undergo fine-tuning on all downstream tasks. Notably, even in this baseline setup, the EC model exhibits performance gains compared to our baseline, reinforcing the efficacy of emergent communication in improving model performance.
> > - Section F: Section F contains a series of qualitative analyses spanning over three pages dedicated to demonstrating examples of the structure and interpretation of EC tokens. This comprehensive analysis delves into the patterns and meanings embedded in EC tokens, providing a nuanced understanding of their properties and how they contribute to vision-language learning.
> >
> > We hope that this overview addresses your concerns regarding the comprehensiveness of our experiments. We believe that the ablation studies and analyses conducted in these sections contribute to a thorough exploration of emergent communication and its impact on vision-language models. Your reconsideration of our work and ratings in light of our responses would be greatly appreciated.

---

> > > ### Author Response · Authors · 2023-11-22
> > > **Continued response**
> > >
> > > **Why the authors choose OFA as the base model for EC?**
> > >
> > > OFA stands out as a state-of-the-art model in cross-modal learning, particularly in vision-language modeling, and has demonstrated leading performance across various benchmark tasks. We selected OFA due to its widespread usage in the vision-language learning community, its well-documented project website, and its clean and accessible code base, which facilitates reproducibility and comparison with other studies.
> > >
> > > We hope that these additional comments help address your concerns and provide a more comprehensive view of our approach. We value your feedback and appreciate the opportunity to enhance the robustness of our research. Your reconsideration of our work and ratings in light of our responses would be greatly appreciated. Thank you for your thoughtful consideration.

---

### Official Review · Reviewer_PktS · 2023-11-01

**Soundness:** 1 poor
**Presentation:** 3 good
**Contribution:** 2 fair
**Rating:** 3
**Confidence:** 4

**Summary:**

This paper presents a methodology for pre-training Vision Language models on images paired with emergent communication (EC) strings prior to fine-tuning on downstream tasks. The EC strings are derived from a speaker model trained for an image reference game task in an emergent communication paradigm (where the speaker and listener agents must converge on a communication protocol).

The main claim of the paper is that pre-training under this paradigm can provide useful inductive biases during pre-training that can help improve performance when fine-tuning on downstream tasks.

As a testbed, the paper uses the architecture of the One For All (OFA) VLM model from Wang et al. 2022, evaluated on three downstream vision-language tasks: Visual Referring Expression (VRE, where the model must generate a bounding box for an object in being referred to by a natural language expression), Visual Question Answering (VQA), and Visual Entailment (VE, where an agent must classify a natural language string as being (1) entailed, (2) contradicted, or (3) neither in relation to a given image).

The core of the experimental results compare three cases to each other:

(Base) A pre-trained OFA model that is not fine-tuned on downstream tasks.
(+EC) A pre-trained OFA model, further pre-trained on a corpus of EC token/image pairs, fine-tuned on downstream tasks.
(+NL) A pre-trained OFA model, further pre-trained on a corpus of natural language / image pairs, fine-tuned on downstream tasks.

On VRE and VE, the +EC model improves over the baseline while achieving lower (or comparable in some cases with more fine-tuning data in the VE task) performance than +NL. In VQA the +EC model is outperformed by the other two variants.

**Strengths:**

* I find the idea of pre-training on emergent communication strings to be compelling, and the paper motivates potential reasons to expect this benefit well (in particular, the idea that the structural properties of a learned EC protocol could yield useful learning signal is intuitive).

* To my knowledge, the proposed methodology of pre-training a VLM model on EC data and the experiments evaluating this are novel.

* The paper is well written, generally clear, and easy to follow.

* The experimental results show promise for the method (however, I have reservations about whether the claims are fully supported by the results, which I've listed under weaknesses).

**Weaknesses:**

My main concern, and the main reason for my ratings, is related to the experimental setup. I am concerned that the presented results do not fully support the conclusions of the paper:

To my understanding the paper argues that EC pre-training may yield benefits for VLM model performance in cases where there may otherwise not be more data containing natural language / image pairs to train on.

With this in mind, I believe the paper could be much stronger with comparison against the following experimental conditions (in addition to the Base, +EC, and +NL conditions already presented):

(1) A Base model that is also fine-tuned on downstream task data, but *not* additionally pre-trained on either EC nor NL data. This would simulate the case where one only has access to (a) the original pre-training data, and (b) the downstream fine-tuning data, and could potentially improve model performance by additionally pre-training on EC data. Without this comparison, I do not believe that it is clear from the presented results if the improved performance of +EC over Base is due to the EC data itself or if it's due to the downstream fine-tuning.

(2) Downstream task performance of a model pre-trained on the original pre-training data as well as EC data, but *not* fine-tuned on downstream tasks. This would simulate the case where one only has the original pre-training dataset and an evaluation set for the downstream task and could potentially improve performance by augmenting pre-training with EC data.

**Questions:**

1. My main question is if there are results available for either of the two conditions I described in Weaknesses? The lack of these results is the main factor affecting my rating.

2. I'm wondering if the token vocabulary used for images is the same between the OFA model and the EC models? Or do they each learn their own independent tokenizer?

---

> ### Author Response · Authors · 2023-11-22
> **Thank you for your insightful review.**
>
> We are glad that you find “the idea of pre-training on emergent communication strings to be compelling” and that “the idea that the structural properties of a learned EC protocol could yield useful learning signal is intuitive.” We have addressed your main concern with experiments showing a comparison against one of the experimental conditions suggested. We also provide answers to your questions below.
>
> **I believe the paper could be much stronger with a comparison against the following experimental conditions (in addition to the Base, +EC, and +NL conditions already presented):**
>
> **A Base model that is also fine-tuned on downstream task data, but not additionally pre-trained on either EC or NL data.**
>
> In response to your suggestion (1), we want to highlight that we have already conducted the experimental condition you proposed, and the results are thoroughly documented in our supplementary section, specifically in section C.1. In this particular setup, our base model undergoes fine-tuning directly on the downstream tasks without additional pre-training on either EC or NL data. It's important to note a subtle but meaningful distinction in this experimental setting. Unlike our main experimental setup where the (+EC and +NL) models were continuously pre-trained on OFA checkpoints, all models in this setting (Base, +EC, and +NL) start from scratch. This eliminates any unfair advantage that any of the models might have had by being pre-trained on top of OFA.
>
> Then, all the models are equally finetuned on the same downstream task, making sure the base model has the same benefit of finetuning (which is what we believe the reviewer is hoping to see). The only difference between the base model and +EC is that while both start from scratch, the +EC model was initially pre-trained from scratch on a collection of +EC data. This design allows us to isolate and assess the pure and unbiased benefits of pretraining on EC data for Vision-Language Models (VLMs).
>
> The results from this experiment are detailed in section C.1, and we observe significant performance improvements for the +EC model over the baseline, particularly in VRE and VQA downstream tasks. Importantly, these improvements affirm that the enhanced performance of +EC over the base model is attributed to the EC data itself, not the additional fine-tuning or exposure to OFA weights.
>
> We kindly invite you to review section C.1 for a more in-depth understanding of our experiment setup and results in this specific paradigm. Your reconsideration of our work and ratings in light of these additional findings would be greatly appreciated. We believe this contributes valuable insights to the discussion on the benefits of emergent communication pretraining in vision-language learning.
>
> **Downstream task performance of a model pre-trained on the original pre-training data as well as EC data, but not fine-tuned on downstream tasks. This would simulate the case where one only has the original pre-training dataset and an evaluation set for the downstream task and could potentially improve performance by augmenting pre-training with EC data.**
>
> Thank you for your suggestion. While we acknowledge the importance of exploring downstream task performance without fine-tuning, simulating scenarios with only the original pre-training dataset as well as EC data, it seems non-trivial to combine EC and NL vocabularies, as they have different structures, semantic nuances, and contextual meanings. We should also note that downstream tasks that we evaluate are in NL format. Adapting a model trained on a combined EC and NL vocabulary to NL-based tasks introduces additional challenges related to language understanding and contextual relevance. We would appreciate any further guidance or insights you might have on how to effectively overcome these challenges and proceed with this experiment. We look forward to your valuable input on this aspect.
>
> **I'm wondering if the OFA and EC models use the same token vocabulary for images, or if they each learn their own independent tokenizer.**
>
> It is important to note that OFA and EC models employ distinct approaches to tokenization. OFA uses byte-pair encoding for linguistic information and unifies this with discretized image encodings in which image patches are encoded and discretized to be used as a part of the unified OFA vocabulary. In contrast, for the EC models, an arbitrary vocabulary size is set, and the token embeddings are learned throughout the emergent communication game. In the referential game, the speaker agent encodes the images it sees and creates a respective image discrete representation for it (namely EC tokens). In short, the token vocabularies are not explicitly shared between the two.

---

> ### Comment · Reviewer_PktS · 2023-11-23
>
> Thank you to the authors for a thoughtful response and for answering my questions.
>
> **C1 Experiments**:
> The results in supplementary section C1 are indeed interesting, and I believe the work could be improved by the inclusion of these results in the main paper. With that said, if I understand correctly the experiments in C1 are using a different experimental setup since the models are trained on a dataset unified across tasks. If my understanding is correct, then that means that this is a different experimental condition than the one from the experiments shown in the main paper and therefore not directly comparable.
> I think the paper could be further strengthened by the inclusion of this experimental condition (i.e. random weights OFA trained from scratch) using the same datasets/splits in the main paper experiments.
>
> **Pre-training on NL and EC data w/out additional fine-tuning**:
> Thank you for the insightful reply, I see your point and I agree with your assessment that running such an experiment would not really be feasible/meaningful given the difficulty introduced by combining the two vocabularies.
>
> Based on the reasons given in my C1 comments, I have decided to retain my original score.

---

### Official Review · Reviewer_2A8R · 2023-11-09

**Soundness:** 2 fair
**Presentation:** 1 poor
**Contribution:** 1 poor
**Rating:** 3
**Confidence:** 4

**Summary:**

This paper explores the use of Emergent Communication (EC) for knowledge transfer in the Vision Language Pretrained Models. It pretrains a model on an EC corpus, then experiments on three tasks, including Visual Referring Expression (VRE) and Visual Entailment (VE). Empirical experiments show that pretrained on EC corpus can improve the performance on the downstream tasks, and highlight the transferability and generalization capabilities of EC pretraining on VL domain.

**Strengths:**

1. Empirical results show the VL models pretrained on EC corpus can be transferred to downstream tasks with improved gain in controlled settings.

**Weaknesses:**

1. Experiment part is pretty weak, lacks of baselines for comparison, especially strong baselines, to verify the effectiveness of the proposed approach.
2. The novelty of the method / approach is also a big concern.

**Questions:**

The major issue of this work is experiment part is too weak, needs more baselines for comparison to show the effectiveness of the approach, given the main approach also lacks of novelty.

---

> ### Author Response · Authors · 2023-11-22
> **Thank you for your thoughtful review**
>
> Please find our responses below:
>
> **Experiment part is pretty weak, lack of baselines for comparison, especially strong baselines, to verify the effectiveness of the proposed approach.**
> In response to this concern, we have introduced an experiment to strengthen our baseline comparisons. Specifically, we modified the experimental setting by enhancing the baseline model. To achieve this, we subjected the baseline model to finetuning on all downstream tasks. This setting eliminates the confounding factor of finetuning in the +EC models, providing a stronger baseline for evaluation. Furthermore, all models in this setting start from scratch, without continuous pre-training on OFA. This ensures that the learned knowledge from OFA's exposure to extensive data does not introduce an inductive bias to our models. Details of this experiment, including results, have been included in supplementary section C.1. It is noteworthy that in this enhanced and stricter baseline setup, our +EC model continues to demonstrate significant performance gains, particularly in VRE and VQA tasks.
>
> **The novelty of the method / approach is also a big concern.** W.r.t. novelty, we would like to emphasize that - to the best of our knowledge - our work is the first to explore the potential benefits of emergent communication pretraining on vision-language learning across several vision-language downstream tasks. We have revised the introduction to reflect our novelty.
>
> Our contribution is important for several reasons:
>
> 1) Emergent communication pretraining involves models developing their own communication protocols to solve tasks. Investigating its impact on vision-language learning helps us understand how models autonomously develop communication strategies, potentially providing insights into the inner workings of the models.
> 2) Assessing the benefits across multiple vision-language downstream tasks explores the transferability of knowledge gained through emergent communication pretraining. This is crucial for developing models that can generalize their learned communication skills to a diverse set of tasks, making them more versatile and effective.
> 3) If emergent communication proves effective for VLM-PTMs, it could be a valuable tool for training models when large labeled parallel vision-language datasets are scarce or expensive to obtain. This has practical implications for real-world applications where resource constraints may be a limiting factor. Exploring emergent communication in this context addresses the unique challenges posed by the integration of visual and linguistic information, potentially leading to future research in this direction.
>
> Notably, its efficacy has been demonstrated in language modeling, where it outperformed other sources of synthetic data, as highlighted by Yao et al. (2022). It is noteworthy that Yao et al.'s work on EC pretraining for NLP tasks received a spotlight in ICLR 2022, indicating the relevance and recognition of emergent communication research within the ICLR community.
>
> We believe our work makes a substantive contribution by extending the exploration of emergent communication to the domain of vision-language learning. We are eager to ensure that your specific concerns are addressed comprehensively. To facilitate this, we kindly request more specific feedback on the aspects of our method or approach that you find lacking in novelty, and any suggestions regarding the baselines you believe our work should incorporate. Your detailed insights will be invaluable in refining and strengthening our work.
>
>
> Thank you once again for dedicating your time and consideration to our submission. We eagerly anticipate receiving more specific feedback from you to further enhance the clarity and contribution of our work. Your reconsideration of our work and ratings in light of these additional findings would be greatly appreciated.
>
>
> **References**
> - Yao et al. (2022) Linking Emergent and Natural Languages via Corpus Transfer. ICLR 2022 Spotlight

---

### Author Response · Authors · 2023-11-22
**Thank all the reviewers for your thorough reviews and insightful suggestions.**

Dear reviewers,

We sincerely appreciate the time and effort each of you dedicated to reviewing our manuscript. We thank reviewers for recognizing the novelty of our work (**PktS**) and its clarity in terms of presentation and explanations (**PktS, DU5f**). Reviewers find our idea of pretraining on EC compelling, well-motivated, and intuitive (**PktS, DU5f**), and our empirical findings clear (**DU5f**), promising (**PktS**), demonstrating that EC pretraning can lead to downstream task improvements (**2A8R**).

We emphasize that our work is the first, to our knowledge, to explore the potential benefits of emergent communication pretraining on vision-language learning across various downstream tasks. We have provided detailed reasoning for the novelty of our contribution. Additionally, we have revisited the introduction to provide a more comprehensive explanation of why EC pretraining is a promising approach for addressing limitations in Vision-Language Models (VLMs). This includes potential applications in low-resource settings, structural and semantic benefits, and past successes in language modeling using emergent communication.

We have conducted several ablation studies and analyses to investigate the components and framework of emergent communication. These studies are detailed in different sections of our supplementary material, providing a comprehensive exploration of emergent communication and its impact on vision-language models.

We genuinely value your feedback and are eager to further refine our work based on your insights. Your reconsideration of our manuscript in light of these additional experiments and explanations would be highly appreciated. Thank you once again for your thoughtful reviews.